# Machine learning method for the prediction of Bedaquiline-resistant *Mycobacterium tuberculosis*

Stuti Ghosh[1], Sudipto Bhattacharjee[2] , Sudipto Saha[1]

The study addresses the increasing resistance to the FDA-approved drug Bedaquiline (BDQ) in *Mycobacterium tuberculosis* (MTB). The absence of any defined resistance locus and the wide variation in the drug targets across clinical isolates have raised a big question about our understanding of the molecular basis of BDQ resistance acquisition. Using machine learning (ML) methods, BDQ resistance was predicted from whole-genome sequencing data for MTB clinical isolates. Variant calling format data generation involved several steps, including adapter trimming and alignment to the H37Rv reference genome. The ML models, namely, Multilayer Perceptron and Random Forest (RF), achieved high accuracies of 83.60% and 79.64%, respectively. The top 50 features were mapped to the H37Rv reference genome, and several new drug targets were identified. In addition to the coding regions, some non-coding intergenic regions were also obtained. Mapping of these features to the H37Rv genome revealed 15 new antibiotic-resistant genes. In addition, the use of explainable AI (XAI) methods, such as SHapley Additive exPlanations, facilitated the identification of mutations associated with BDQ resistance. In conclusion, the ML models demonstrated effective predictive capabilities for BDQ resistance, whereas XAI contributed to understanding key resistance features.

## Introduction

The problem of drug resistance in tuberculosis (TB) poses a significant threat to human health globally. The group A medications for the treatment of drug-resistant tuberculosis include the drugs levofloxacin or moxifloxacin, bedaquiline (BDQ), and linezolid. BDQ is a newer drug recommended by the WHO for treating drug-resistant tuberculosis (Walker et al, 2022). BDQ got FDA approval for the treatment of drug-resistant tuberculosis in the year 2012 (Mahajan, 2013). Another drug in choice for the treatment of drug-resistant TB is clofazimine (CFZ), initially used for the treatment of leprosy (Stadler et al, 2023). In January 2021, the WHO revised the definitions of pre-extensively drug-resistant (pre-XDR) and extensively drug-resistant (XDR) TB. Pre-XDR TB is now defined as the condition where the *Mycobacterium tuberculosis* strain is resistant to any FLQ drug, in addition to fulfilling the condition for MDR/Rifampicin resistance. According to the new convention, XDR TB is the condition in which the infecting *M. tuberculosis* (MTB) strain is, in addition, resistant to at least one group A drug. Delamanid (DLM) is a drug used in the treatment of multidrug-resistant TB (MDR-TB), which got approval from the European Medicines Agency (EMA) in 2013 (Xavier & Lakshmanan, 2014). The rising incidence of bedaquiline-resistant strains presents a severe challenge to the global health community in managing tuberculosis. The rapid emergence of resistance to BDQ plays a crucial role in the transition from pre-XDR to XDR TB condition (Chesov et al, 2022). The drug BDQ was first discovered in 2005 by Andries et al as a diarylquinoline compound R207910 and was explicitly found to target the proton-pumping mechanism of ATP synthase (Andries et al, 2005). Whereas targeting the *atpE* gene comes under the targeted mechanism for the activity of the drug compound, there are some non-targeted mechanisms for its activity. Mutations in the *Rv0678* gene, which codes for a repressor of the mmpS5-mmpL5 efflux pump transcriptional activity, are the most critical among the non-targeted mechanisms (Andries et al, 2014). Mutations were reported in several other target genes, such as *atpE*, *pepQ*, *Rv0678* (Almeida et al, 2016), *Rv1979c*, *Rv3696c*, *Rv2535c*, *Rv0676c*, *Rv0677c*, and *Rv2082* (Hu et al, 2023). A mutation in the target gene *Rv1453* was occurring in cases of BDQ-CFZ cross-resistance, which is also a significant issue, making the situation even worse (Shang et al, 2023). In addition, the patterns in the mutation locus are complex and were found to vary from one clinical isolate to another. In addition, finding mutations in the non-target genes raises a big question about our knowledge of the BDQ antibiotic-resistance genes (ARGs) (Deshkar & Shirure, 2022). With no new drugs available for the treatment of XDR TB, there is an urgent need to understand the molecular mechanism behind the development of resistance to this novel drug.

Although phenotypic drug susceptibility testing (DST) is still the most widely used approach for detecting resistance in MTB, treatment effectiveness is seriously threatened by the delay resulting from the slow growth of the bacteria. The process of whole-genome sequencing (WGS) has emerged as a convenient

---

[1]Department of Biological Sciences, Bose Institute, Kolkata, India    [2]Department of Computer Science and Engineering, University of Calcutta, Kolkata, India

Correspondence: ssaha4@jcbose.ac.in, ssaha4@gmail.com

and rapid diagnostic method for the detection of MDR-TB (Coll et al, 2015). However, genome sequencing data analysis poses a barrier to the general use of WGS technology in clinical tuberculosis because it necessitates bioinformatics knowledge and high-performance computing, which are not readily available in most clinical laboratories (Yang et al, 2018). To analyse MTB WGS data, several tools were developed recently: KvarQ, PhyResSE, TGS-TB, CASTB, Mykrobe, TBProfiler, MTBseq, and ReSeqTB-UVP (Steiner et al, 2014; Bradley et al, 2015; Feuerriegel et al, 2015; Iwai et al, 2015; Sekizuka et al, 2015; Kohl et al, 2018; Phelan et al, 2019; Saluzzo & Maria Cirillo, 2023). However, no such tools can be used for BDQ resistance prediction. Recently, machine learning (ML) approaches have shown promising results in faster detection of drug resistance, specifically against single drugs in the case of MTB. More studies are needed on applying machine learning algorithms to detect MDR and pre-XDR TB (Nimmo et al, 2024).

Considering the severity of the current issue, the application of machine learning approaches can be a potential option in determining BDQ resistance (Sekizuka et al, 2015). Here, we propose an ML-based approach for determining BDQ resistance from publicly available WGS datasets. Furthermore, we have expanded our work to develop a web-based tool called BDQR-MTB and a standalone tool called BDQR-MTB-standalone, useful for the prediction of BDQ resistance for the MTB clinical isolates with the WGS sequences given as input, either as raw data (FASTQ format) or processed data (VCF format).

# Results

## Description of the datasets

WGS data were obtained from 632 MTB clinical isolates across 17 BioProjects IDs. This included 294 BDQ-resistant and 338 BDQ-susceptible isolates. The number of samples considered for model training using fivefold cross-validation were 507 and those for blind testing were 125. The distribution of the resistant and susceptible samples considered for this study is provided in Fig 1A. Of the 294 resistant MTB clinical isolates considered for the current study, 200 were found to be exclusively resistant to BDQ. Some of the BDQ-resistant samples were also found to be resistant to other anti-TB drugs, CFZ (n = 27), linezolid (n = 61). In addition, two of the BDQ-resistant samples were found to be resistant to another anti-TB drug, DLM. On the other hand, most of the susceptible samples (n = 308) were found to be multidrug-resistant (MDR), that is, showing resistance to other drugs such as isoniazid, rifampicin, ethambutol, and fluoroquinolones.

## Performances shown by the models trained using different ML algorithms

The input feature vectors for the machine learning models consisted of 8,282 mutations. Four different read depth thresholds were used to evaluate the performance of the ML models, as shown in Table S1, which indicated that a read depth of 150 was optimal based on sensitivity and computational cost.

The optimal Random Forest (RF) and Multilayer Perceptron (MLP) models achieved accuracies of 79.64% and 83.60%, respectively, using fivefold cross-validation. The sensitivity and specificity metrics shown by the RF model were 85.92% and 74.28% whereas those shown by the MLP model were 86.31% and 81.26% during fivefold cross-validation. Both the optimal MLP and RF models achieved the same area under the curve (AUC) of 0.87. The optimal RF and MLP models achieved accuracies of 81% and 84% during testing on the blind dataset. The performance of the optimal MLP model was further evaluated on an independent dataset comprising samples from different countries. The results including the ML model metrics obtained from this testing is shown in the Table S2. On the other hand, the ML model trained with the Support Vector Machine (SVM) algorithm achieved accuracy, sensitivity and specificity metrics of 73.78%, 63.25%, and 82.84% during fivefold cross-validation and 72.80%, 56.89%, and 86.56% during blind testing. The SVM model achieved an AUC of 78% during fivefold cross-validation. The performance measures for the ML models trained with RF, MLP, and SVM algorithms, both during training using fivefold cross-validation and blind testing, are depicted in Table 1. Threshold tuning was performed for the MLP, RF, and SVM models to obtain the performance metrics at various thresholds. A threshold determines the decision boundary for the classification task of ML models. The models were found to be performing optimally at thresholds of 0.8 for the MLP, 0.6 for the RF model and 0.5 for SVM. The results of threshold tuning for the trained MLP, RF and SVM models are provided in Table S3. Hyperparameter tuning was performed to obtain the optimal MLP and RF models. For the RF model, hyperparameter tuning was performed by varying the mtry levels. On the other hand, for both the single and MLP models, hyperparameter tuning was performed by using different node combinations. Fig S1 shows the ROC curves and AUC values for the hyperparameter tuning of RF, single-layered and MLP models. The model architecture for the optimal MLP model training is provided in Fig S2. The bar diagrams depicting threshold-dependent metrics, that is, accuracy, sensitivity and specificity of the RF, MLP and SVM models, are shown in Fig 1B. The Receiver Operating Characteristics (ROC) plots and the corresponding AUC values are shown in Fig 1C. The ROC plots showing the AUC values for the ML models trained with different numbers of top contributing features that is top 5, top 10, top 20, top 50, top 100, and top 200 features selected from the optimal MLP model, are depicted in the Fig 1D. The optimum performance was found to be shown by the MLP model trained with the top 50 features (AUC: 0.78). So, the MLP model trained with the top 50 features was taken as the minimalistic model.

## Model explainability using SHAP

The top 50 resistance features (gene mutations) identified by both optimally performing MLP and RF models were ranked by SHAP importance scores. A representative box plot of the AO/DP value distribution for the top 50 features selected from the optimal RF model across BDQ-resistant and susceptible samples is shown in Fig 2A. The AO/DP value distribution for the top 50 features selected by the optimal RF model was found to more

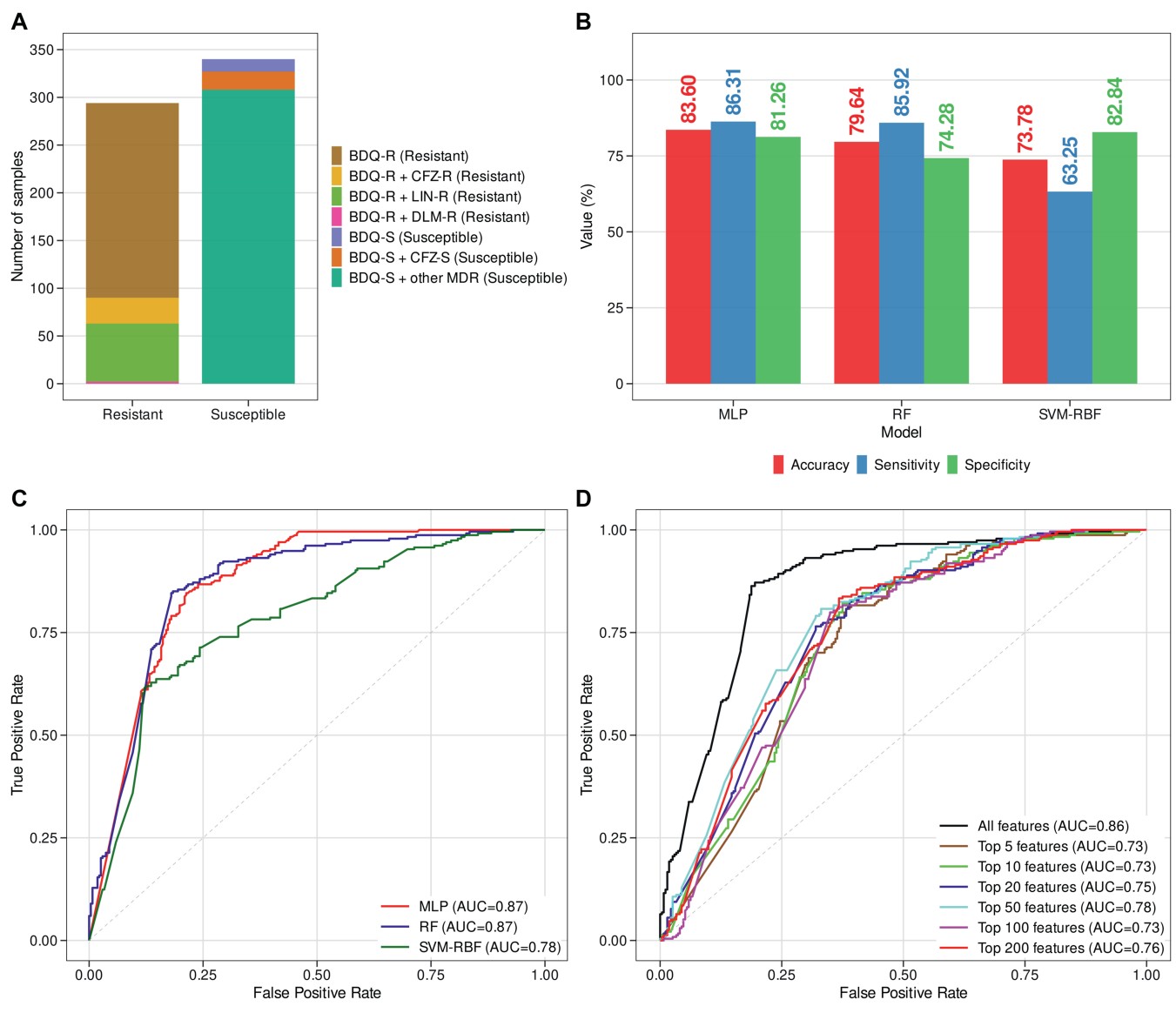

**Figure 1. The dataset used for ML and performance metrics of the models.**
**(A)** The distribution of the dataset showing the different types of BDQ-R and BDQ-S MTB clinical isolates. **(B)** The threshold-dependent metrics accuracy, sensitivity, and specificity obtained on training of resistant and susceptible samples using the fivefold cross-validation technique. The ML models were trained with different algorithms, such as Multilayer Perceptron (MLP), Random Forest (RF), and Support Vector Machine. **(C)** The threshold independent metric AUC or area under the curve was obtained on ML model training using different ML algorithms ML model training using different numbers of top contributing features, selected based on the optimally performing MLP model. **(D)** ML model training was performed using different numbers of top contributing features, selected based on the optimally performing MLP model. The ML model trained on the top 50 features achieved the best performance (AUC: 0.78).

clearly differentiate between BDQ-resistant and susceptible isolates. Box plots of the AO/DP value distribution for the top 50 features selected by the optimal MLP model are depicted in Fig S3. The union set from the Venn diagram of top$_{50}$RF_BDQR$^{high}$ and top$_{50}$MLP_BDQR$^{high}$ the top 50 features selected from both the optimal MLP and RF models and with AO/DP value > 0.7 for the resistant isolates, were found to represent a total of 62 features, where seven features, namely A4120983G, G21795A, A1481185C, C1481563T, GGATG2338990CGATA, T283614C, and C623508G, were found in the intersection set. The total list of the above-mentioned 62 features is provided in Table S4. The Venn

diagram of top$_{50}$RF_BDQR$^{high}$ and top$_{50}$MLP_BDQR$^{high}$ is given in Fig S4. In the next step, the proportion of BDQ-resistant MTB clinical isolates with these features was computed. The details on the number of features obtained from the optimal MLP and RF models, and the proportion (%) of their occurrences among BDQ-resistant samples, are depicted in Table S5. The number of features present in 5%, 10%, 15%, and 20% of the resistant isolates was checked. An optimal set of features, comprising 10 features from each of the optimally performing MLP and RF models, was found to be present in 15% of the BDQ-resistant samples considered for this study (n = 294). Thus, this set of

**Table 1. The performance metrics for the different machine learning algorithms during training with fivefold cross-validation and testing on the blind dataset.**

| Machine learning algorithm | Training with fivefold cross-validation (n = 507) | | | | Testing using blind dataset (n = 125) | | | |
|---|---|---|---|---|---|---|---|---|
| | Accuracy (%) | Sensitivity (%) | Specificity (%) | AUC (%) | Accuracy (%) | Sensitivity (%) | Specificity (%) | AUC (%) |
| Random Forest (mtry = 3,471) threshold: 0.6 | 79.64 | 85.92 | 74.28 | 87 | 80 | 82.76 | 79.1 | 87.18 |
| Support Vector Machine with radial basis function (Sigma-Aldrich = 0.27, C = 1) (threshold: 0.5) | 73.78 | 63.25 | 82.84 | 76.35 | 72.8 | 56.89 | 86.56 | 82.2 |
| Multilayer Perceptron (layer1 = 130) (threshold: 0.8) | 83.6 | 88.89 | 81.26 | 87 | 84 | 82.75 | 85.07 | 87.36 |

features was considered for further evaluation. A Venn diagram was performed with top$_{50}$RF_BDQR$^{high}$_15% and top$_{50}$-MLP_BDQR$^{high}$_15% the features selected from RF and MLP models, which were present among the top 15% proportion of the BDQ-resistant MTB clinical isolates (Fig S5). The union set from the Venn diagram of top$_{50}$RF_BDQR$^{high}$_15% and top$_{50}$-MLP_BDQR$^{high}$_15%, showed a total of 19 features, whereas the intersection set consisted of one feature, A4120983G (intergenic region, upstream of *whiB4*). These top 19 features were mapped to the H37Rv reference genome, leading to the identification of 15 ARGs (two already known and 13 novel genes) determined to be potentially important for BDQ resistance. In addition, four different intergenic regions were also identified. The circular genome diagram of H37Rv, showing the mapping of these 15 identified potential ARGs and the 4 intergenic regions obtained, is depicted in Fig 2B. The genome diagram was predicted using the Proksee server (Grant et al, 2023) (https://proksee.ca/). A genome diagram is particularly useful in representing the precise chromosomal locations of genes. Among the identified targets, 2 ARGs, namely, *mmpL5* and *Rv2082*, are known for BDQ drug resistance (Hu et al, 2023). The features mapping to intergenic regions were identified as "A4120983G," "TA3336679T," "G3232759A," and "C39030T." The details of the 19 selected top features for BDQ resistance, that is, the target genes and intergenic regions obtained by mapping the top features against the H37Rv reference genome, are shown in Table S6. The feature "A4120983G," found to be upstream of *whiB4*, was recognised as a highly contributing feature obtained from both the optimal MLP and the RF models. The chromosomal locations of the four intergenic regions and their upstream and downstream genes are shown in Table S7. A promoter region was observed within ~60 bp from the intergenic locus position 4,120,983, as shown in Table S1. The numbers of BDQ-resistant and susceptible MTB clinical isolates with features in the known target genes for BDQ resistance are depicted in Fig 2C. The representative domain groups identified in the encoded proteins from the ARGs in our study and their respective hypergeometric probabilities are shown in Table S8. The significance of the domains was categorised based on the obtained *P*-values. Certain domain groups, which were found to be enriched among the BDQ-resistant MTB clinical isolates, included transmembrane domains, the arabinan synthase domain, and domain of unknown function (DUFs). The model

trained with the top 50 features was explained using the SHAP algorithm.

### The utility of the BDQR-MTB web application

The BDQR-MTB homepage provides the facility to upload an input VCF file and also contains a button to get an example VCF file that can be used for demonstrative purposes. The snapshots of the homepage and the output page of the BDQR-MTB web application are depicted in Fig 3A and B. The output page shows the Sample ID of the input sample, the prediction result (susceptible or resistance), and the prediction probabilities as computed using the full model. It also shows the top 20 contributing mutations and their gene mappings as computed by SHAP on the minimal model. A web-based application, named BDQR-MTB, was developed, for which the saved models were stored on a web server. BDQR-MTB is freely available at http://bicresources.jcbose.ac.in/ssaha4/bdqr-mtb. Users can access BDQR-MTB using only a web browser with internet connectivity. A standalone version called BDQR-MTB standalone was also developed. This is a command-line application which can be used on any Linux operating system. BDQR-MTB-standalone can be freely downloaded from https://github.com/PulmonomicsLab/bdqr-mtb-standalone. It provides the facility to predict the BDQ resistance of (i) a single sample in FASTQ files, (ii) multiple samples in FASTQ files, (iii) a single sample in a VCF file, (iv) multiple samples in multiple VCF files, and (v) multiple samples in a single (merged) VCF file. The workflow of the BDQR-MTB standalone is shown in Fig 3C. The SHAP plot showing the top 10 contributing features for a BDQ-resistant sample with the run ID ERR978256 is given in Fig 3D.

## Discussion

The lack of a well-defined genetic hotspot for BDQ resistance in MTB clinical isolates has put a significant blockage to the process of predicting resistance against this drug (Lange et al, 2023). In this study, we have used WGS data of BDQ-resistant and susceptible MTB clinical isolates to predict BDQ resistance using ML-based approaches. The top 50 features contributing to BDQ resistance prediction were determined using both RF and MLP, as both performed optimally across specificity, sensitivity, and the

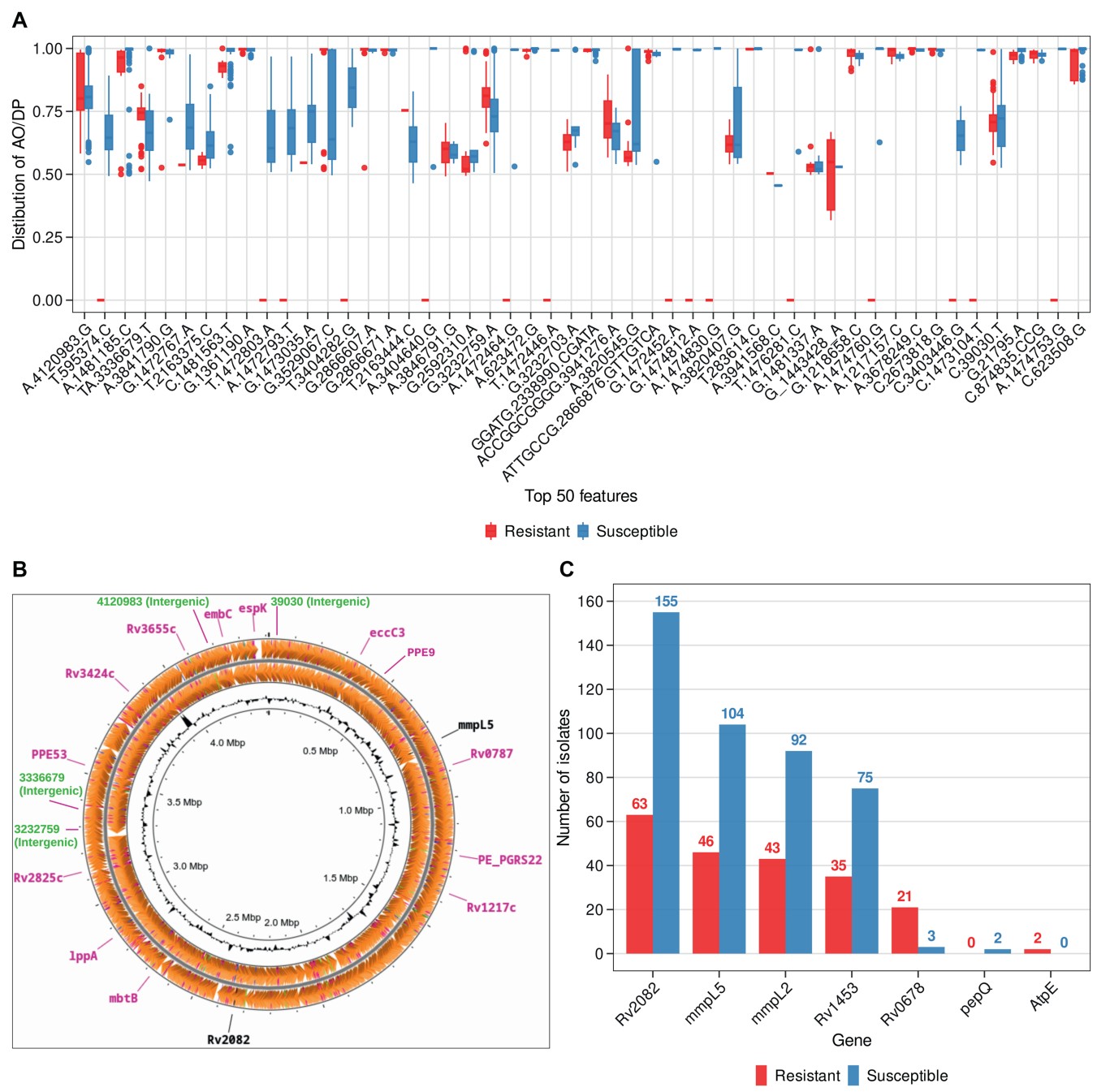

**Figure 2. Frequency analysis and MTB genome map of the RF model–derived top contributing features linked to BDQ resistance.**
**(A)** Box plot showing the distribution of AO/DP values for the top 50 features obtained based on the optimal RF model across BDQ-resistant and susceptible isolates.
**(B)** The circular genome diagram obtained from mapping the top RF and MLP features (AO/DP > 0.7) to the H37RV reference genome. The known ARGs were shown in black, and the novel ones in pink. The chromosomal locus mapping to the intergenic regions was represented in green. **(C)** The frequencies of BDQ-resistant and susceptible isolates with mutations in the already known target genes for BDQ were obtained and represented as a bar diagram.

threshold-independent AUC metric. In this study, the positive class denoted BDQ resistance. Thus, the metric sensitivity, which is the ML model's efficiency in correctly predicting the positive class, played an essential role in determining the resistant class. The AO/DP distribution for the top 50 features selected from the RF model was more efficient in differentiating between BDQ-resistant and

susceptible samples. Thus, the features selected by the RF model were estimated to be more biologically significant. The top features selected by the RF and MLP models were filtered to select the set of features for which the AO/DP values were higher in resistant samples (greater than 0.7) than in susceptible ones. Further selection of features significantly present among the top 15% of the

**A**

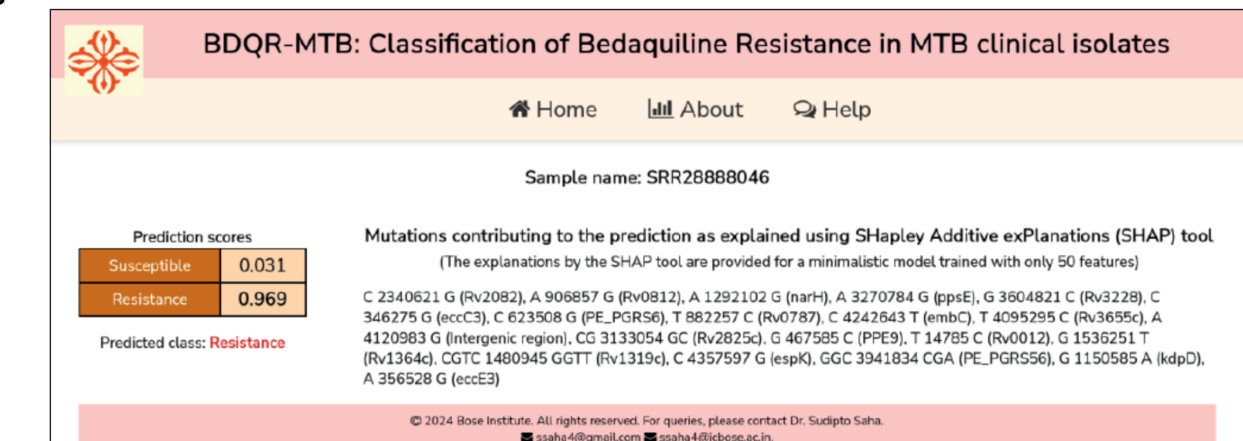

**B**

**C** **D**

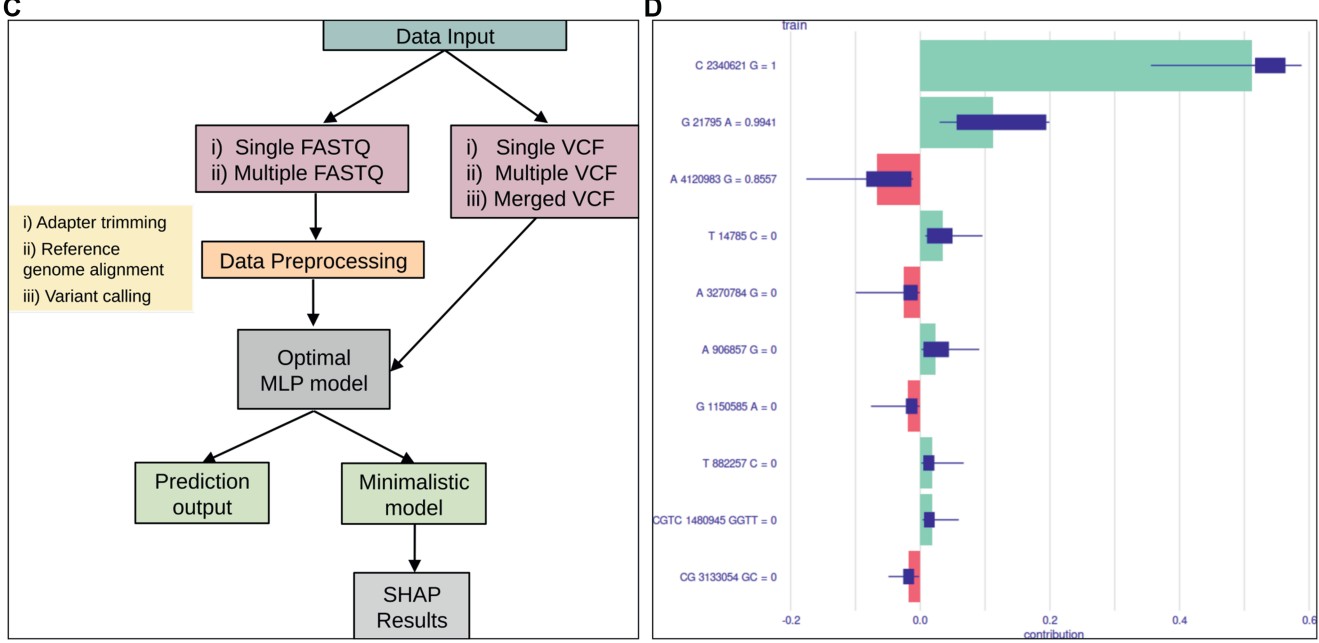

**Figure 3.  BDQR-MTB web and standalone version applications.**
**(A)** The homepage for the BDQR-MTB web application, where a VCF file can be given as an input. **(B)** The output page showing the results obtained from the BDQR-MTB web application. **(C)** The workflow for the BDQR-MTB standalone version, where the FASTQ and VCF format files for either a single sample or multiple samples can be given as input. **(D)** The SHAP plot for a BDQ-resistant sample showing the top 10 features that contribute mostly to the BDQ resistance prediction in MLP model. Green

BDQ-resistant samples led to the identification of 13 novel ARGs necessary for BDQ resistance. Two well-known ARGs for BDQ resistance, namely, *mmpL5* and *Rv2082*, were also identified. In addition to the above, four different intergenic regions were also identified. The finding of a promoter region with ~60 bp from locus position 4,120,983 (mapping to the intergenic region) further puts importance on the feature "A4120983G," which was obtained as the topmost contributing feature from both the optimal MLP and RF models. This intergenic locus was found to be upstream of *whiB4*, which is a transcriptional regulatory protein. The gene *whiB4* was reported to function as a transcriptional regulator, showing a response to the binding of oxygen and nitric oxide (Chawla et al, 2012). In addition to the above, several other known drug-resistant genes such as *atpE*, *Rv0678*, *pepQ*, *Rv1979c*, and *Rv1453* were also obtained in our study. Furthermore, the application of explainable AI using the SHAP algorithm was used to determine the top 10 contributing features to BDQ resistance. This study further investigated the functional categories of BDQR ARGs in MTB, examining the representative domains and their functional categories. Hypergeometric probability analysis helped identify domains of significance. The transmembrane domain, arabinan synthase domain, and DUF were found to be enriched in the BDQ-resistant MTB clinical isolates.

Our study trained the ML model exclusively to predict BDQ resistance. In fact, our study was the first to show the application of ML to identify novel ARGs, focusing solely on the MTB clinical isolates. One study in 2019 focused on the ML-based identification of BDQ resistance-associated mutations found exclusively in the *atpE* gene, taking data from both in vitro and clinical studies (Karmakar et al, 2019). There are several other ML studies to determine resistance against other anti-TB drugs like pyrazinamide, ethambutol, streptomycin, capreomycin, isoniazid, and fluoroquinolones. Green et al and Chen et al used convolutional neural network (CNN)-based ML models to predict anti-TB drug resistance from MTB clinical isolates, outperforming other algorithms such as RF and SVM (Chen et al, 2019; Green et al, 2022). Our web server BDQR-MTB is the first one to solely focus on BDQ resistance prediction and has shown the application of explainable AI to determine the top contributing features.

In clinical settings, some BDQ-resistant MTB clinical isolates were found to show no mutations in *Rv0678*, *atpE*, or *pepQ*, even though these mutations impart significant resistance to BDQ, indicating the existence of additional unidentified mechanisms of resistance to BDQ (Andres et al, 2020). Our study also shows the higher frequencies of mutations in the known target genes of BDQ resistance among BDQ susceptible MTB isolates as compared with the resistant ones (Fig 2C). Moreover, there is considerable variability in the mutation patterns between the MTB clinical isolates and the in vitro selected mutants of BDQ (Sonnenkalb et al, 2023). To focus our interest solely on the MTB clinical isolates, the in vitro mutants were excluded from our considerations in the current study. In addition to the above, the RAVs present in the above-mentioned ARGs, such as *atpE* and *pepQ*, are more likely to be present in in vitro–selected mutants (Das et al, 2021). Another study indicated the absence of mutations in the known target genes *Rv0678* and *pepQ* in the case of the BDQ-naïve population and the complete absence of *atpE* mutations from the clinical isolates of BDQ (Derendinger et al, 2023). Nguyen et al indicated the absence of *atpE* mutations in around 72% of the isolates (Nguyen et al, 2018). This can give a possible explanation for the low frequencies of the known target genes among the sequenced isolates and the subsequent absence of some of them among the top 50 features.

Several studies point to the roles of the transmembrane domain, arabinan synthase domain and DUF as obtained from our study in resistance against different anti-TB drugs. The transmembrane domains in the *ubiA* gene were found to harbour most of the mutations in cases of ethambutol resistance (Lingaraju et al, 2016). DUF will likely acquire resistance against aminoglycosides such as kanamycin and capreomycin (Sharma et al, 2016). PE/PPE is a conserved protein group in Mycobacterium that influences host immune responses, defence mechanisms, and cell fates, which are crucial for MTB survival (D'Souza et al, 2023).

The BDQ-susceptible MTB clinical isolates may show resistance against the other anti-TB drugs. On the other hand, BDQ-resistant MTB isolates may also be resistant to additional anti-TB drugs such as CFZ, as cross-resistance between BDQ and CFZ is very prevalent (Fig 1A). Thus, our study's top 50 features may also be involved in resistance against other anti-TB drugs. This explains why our model is unable to accurately predict the remaining 18% of the samples. However, we have exclusively selected the top 50 genes (features) important for BDQ resistance based on our optimally performing trained MLP model.

In summary, in this study, we have shown the application of ML-based approaches for the prediction of BDQ drug resistance and using explainable AI, we could determine the top features contributing mostly to BDQ resistance acquisition.

## Materials and Methods

### Data curation

WGS files in FASTQ format were obtained from the European Nucleotide Archive (ENA) for the BDQ-resistant and the susceptible MTB clinical isolates. A total of 632 MTB clinical isolates (294 BDQ-resistant and 338 BDQ-susceptible) from 17 different BioProjects IDs were considered for this study. The dataset was selected using specific inclusion and exclusion criteria. This study used only paired-end WGS data from clinical isolates of MTB. Prerequisites for the eligible datasets are (i) BDQ-associated phenotypic DST results and (ii) accessible metadata, such as sample origin and sequencing information when available. As prior research indicates significant variation in mutation patterns between BDQ-resistant MTB clinical isolates and the in vitro–selected BDQ mutants, datasets generated from non-clinical or laboratory

---

bars indicate feature contributions toward resistance prediction, red bars toward susceptibility, and blue bars represent the distribution of SHAP values across perturbations for each feature.

strains lacking phenotypic DST data were excluded (Rivière et al, 2022; Derendinger et al, 2023). Furthermore, datasets with inadequate coverage or poor sequencing quality, as well as single-end sequencing data, were not included in the analysis.

## Data preprocessing and variant calling

The WGS reads were trimmed using TrimGalore (version: 0.6.7) and mapped to the H37Rv MTB reference genome (RefSeq: NC_000962.3) using BWA-MEM (version: 0.7.17-r1188), to generate the BAM files (Li, 2013 Preprint; Krueger et al, 2021). The BAM files were processed using the samtools (version: 1.13) (Li et al, 2009). Variant calling was performed using freebayes (version: v1.3.6) (Garrison & Marth, 2012 Preprint). The tool vcflib (version: 1.0.7) was used to filter the variant calling format (VCF) files (Müller et al, 2017) based on read depth (DP) and read quality (QUAL) (DP>150 and QUAL>30). To select the DP cutoff value, the model performances were evaluated with datasets obtained by the application of different DP filtering cutoffs. The filtered VCF files were merged using the bcftools (version: 1.22) (Danecek et al, 2021), and the merged VCF file was obtained. The alternate allele ratio was determined for each mutation event by applying the formula:

$$\text{Alternate allele ratio} = AO/DP.$$

where read depth (DP) and alternate allele observation count (AO) are expressed. Each mutation event was considered a feature for training and testing the ML algorithms, and the alternate allele ratio was considered the value for the input feature vector. For BDQ-R, the samples were labelled as R, and for BDQ-S, as S. The details of the BioProject IDs are provided in Table S9. The run IDs and category labels for the 632 samples in our dataset are provided in Table S10. The VCF files provide information about the mutational events, including the single-nucleotide polymorphisms and the insertion-deletions (INDELS).

## Training and blind testing dataset construction

The dataset was randomly split in 80:20 proportion. The 80% data were used for machine learning (ML) model training using fivefold cross-validation. The remaining 20% data were used for blind testing.

## Machine learning pipeline and SHAP-based explainability for BDQ resistance prediction

The merged VCF file obtained from the WGS reads of the MTB clinical isolates was given as the input for the process of ML model development. The caret package in R was used to split data, tune ML models, and train and test to evaluate the models (Kuhn, 2008). Three classic machine learning classifiers: RF (Breiman, 2001), SVM (Zhang, 2012) with Radial Basis Function (RBF) kernel, and MLP (Murtagh, 1991) were trained using fivefold cross-validation with the training and testing datasets. The RF algorithm is based on decision trees, where the final outcome of the model is an aggregation of the individual trees (Breiman, 2001). The SVM algorithm involves identifying an optimal decision boundary, referred

to as the "hyperplane," to differentiate between the different classes (Zhang, 2012). The MLP algorithm involves a fully connected neural network-based model, consisting of an input layer, an output layer and one or more hidden layers (Sarker, 2021). To optimize the performance of the ML models threshold tuning and hyperparameter optimisation were performed. Threshold-dependent metrics such as accuracy, sensitivity, and specificity, as well as the threshold-independent metric AUC (ROC curve) were used for the evaluation of the ML model performances. The top contributing features were selected by using the VarImp function and MLP models were trained with top 5, top 10, top 20, top 50, top 100, and top 200 contributing features. The MLP model trained with top 50 features was obtained as the minimalistic model. Explainable AI (XAI) called SHAP was applied for the determination of model explainability. The minimalistic model and the explainer model were used for the development of the BDQR-MTB web application and the BDQR-MTB standalone versions. A detailed overview of the methodologies for the different steps: data curation of the paired WGS reads in FASTQ format for the BDQ-resistant and susceptible MTB clinical isolates, processing of the FASTQ files to obtain the VCF format files, and machine learning, followed by explainable AI, is depicted in Fig 4A–C.

## Model evaluation

The sample class was predicted based on the highest prediction probability for each sample. Threshold tuning was performed on the chosen optimal model to evaluate its performance at different thresholds. Different threshold-dependent metrics, such as accuracy, sensitivity, and specificity, were used to select the best threshold for optimal model performance.

## ROC-AUC as a threshold-independent measure of model performance

The receiver operating characteristics (ROC) curve measures the classification ability of a machine learning model at different threshold settings. The AUC value measures the classification performance (Bradley, 1997). The ROC curves were plotted to evaluate the performance of models trained with different ML algorithms such as RF, SVM-RBF, and MLP: single-layered MLP models trained with varying numbers of nodes, MLP models trained with different node combinations, and MLP models trained with differing numbers of top contributing features.

## Explainable AI

The variable importance scores for the top 50 variables (mutations) were calculated based on both the optimal MLP and the RF models. Both the RF and the MLP models were considered for selecting the top 50 features, as both showed optimal performances based on the threshold-dependent metric accuracy, sensitivity and threshold-independent metric AUC. The top 50 features selected from the RF model were named as $top_{50}RF$, and those selected from the MLP were represented as $top_{50}MLP$. Furthermore, of the top features selected from both the optimal ML models, those with the AO/DP values greater than 0.7 for the BDQ-

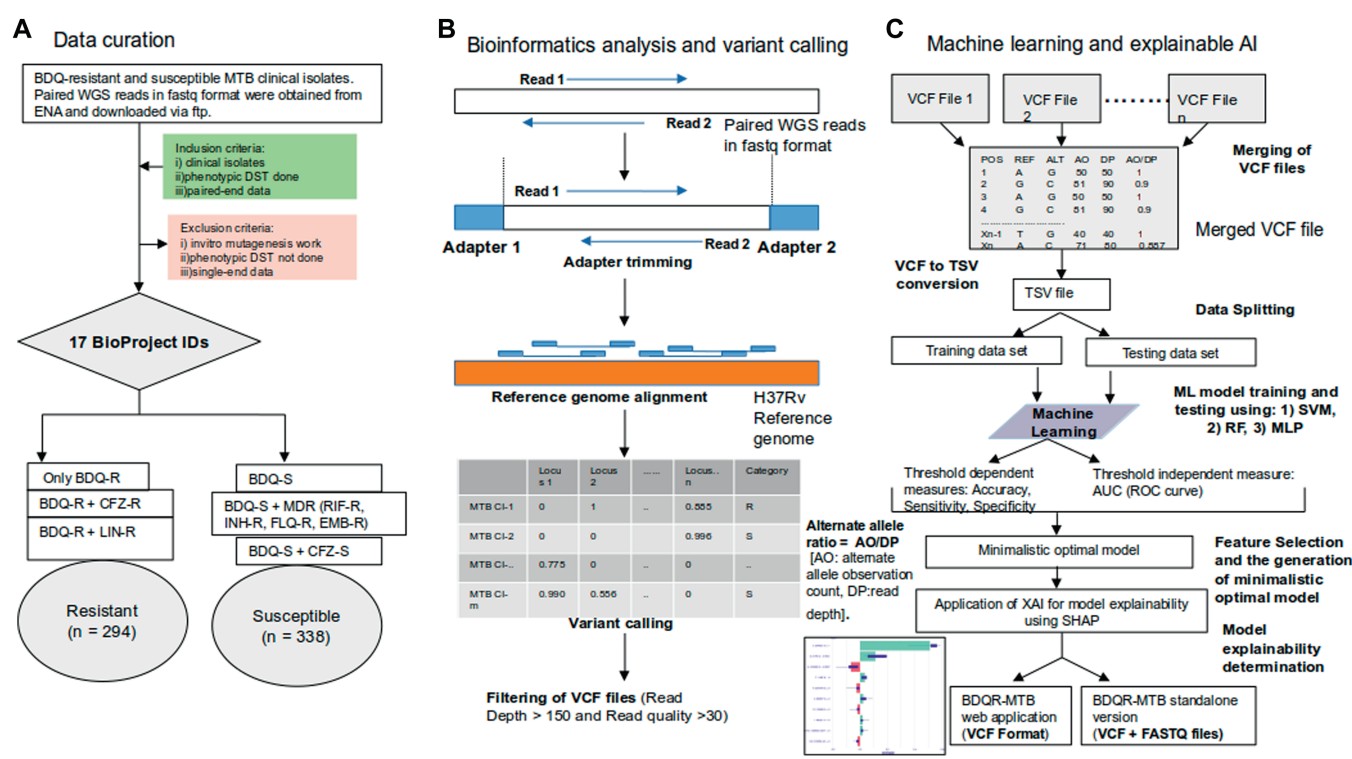

**Figure 4. The overall workflow of the study, consisting of three steps: data curation, bioinformatics analysis, and machine learning.**
**(A)** WGS data of MTB clinical isolates for BDQ-resistant and susceptible were selected based on inclusion and exclusion criteria. **(B)** The steps of bioinformatics analysis, including adapter trimming, alignment to the *H37Rv* reference genome, variant calling, and filtering to generate high-quality VCF files. **(C)** Merged VCF files were used as features in the machine learning workflow, feature selection, and explainability for BDQ resistance prediction.

resistant samples were determined. These feature sets from both models are represented as $top_{50}RF\_BDQR^{high}$ and $top_{50}MLP\_BDQR^{high}$. A Venn diagram was plotted with $top_{50}RF\_BDQR^{high}$ and $top_{50}MLP\_BDQR^{high}$ to obtain a union set. Next, the number of features present among different proportions, 5%, 10%, 15%, and 20% of the BDQ-resistant MTB clinical isolates, were computed (n = 294). The following part consisted of selecting those features of this list that were found to be present among the top 15% of the BDQ-resistant MTB clinical isolates. These feature sets were named as $top_{50}RF\_BDQR^{high}\_15\%$ and $top_{50}MLP\_BDQR^{high}\_15\%$ for those selected from either the RF model or the MLP model. Another Venn diagram was drawn with the sets $top_{50}RF\_BDQR^{high}\_15\%$ and $top_{50}MLP\_BDQR^{high}\_15\%$. The features obtained from the union of these two sets were now mapped to the H37Rv reference genome, and a genome diagram was drawn with the mapped target genes and the intergenic regions obtained. The intergenic regions were further studied to obtain the locations of the promoter regions for the downstream genes. The promoter finding tool BPROM (http://www.softberry.com/berry.phtml) was used to obtain information about the promoter regions. The minimal optimal model was obtained by training with the top 50 variables, and the model's performance was evaluated. Furthermore, their functional categories were determined, followed by an extensive study of the representative domains, and the hypergeometric probability of the domains from proteins of each functional category was determined (https://stattrek.com/online-calculator/hypergeometric).

The contribution scores for each variable were computed based on the SHAP algorithm, and a list of probable mutations was obtained. The DALEX package in R was used for SHAP (Baniecki et al, 2020 *Preprint*).

## Web-based and standalone applications

A web-based application, named BDQR-MTB, was developed, for which the saved models were stored on a web server. BDQR-MTB is freely available at http://bicresources.jcbose.ac.in/ssaha4/bdqr-mtb. The trained optimal models were saved as files. The models include the optimal full model (trained with 8,282 features), the minimal model (trained with 50 features), and the SHAP explainer model. The models were saved as binary files native to the R programming language using R Data Serialisation (RDS) format. The home page contains a web form which provides the user with input facilities. The form can be used to upload a Variant Call Format (VCF) file containing the set of input mutations of a sample. The home page was written using HyperText Markup Language (HTML), Cascading Style Sheets (CSS), and JavaScript. The server-side scripts were written using PHP (PHP: Hypertext Processor [recursive acronym]), R, and shell script languages. The server-side PHP script first receives the uploaded VCF file, performs initial validations, and transfers the VCF file to a shell script. The shell script generates a vector of mutations in a tab-separated values (TSV) format file using the vcflib software and calls an R script with

the TSV file as an argument. The R script loads the saved models from the RDS files, performs the BDQ resistance prediction of the input sample (TSV file) using the saved models, and returns the result to the PHP script in JavaScript Object Notation (JSON) format. Then, the PHP script sends the output back to the client-side as an HTML output page. The workflow for the BDQR-MTB web application is provided in Fig S6. A serverless standalone application named BDQR-MTB-standalone (https://github.com/PulmonomicsLab/bdqr-mtb-standalone) was also developed, and saved models were packaged with the application. The standalone version was implemented using R and shell scripting languages. For BDQ resistance prediction from FASTQ files, shell scripts initially execute a pipeline to generate the VCF files, which include processing the FASTQ files using TrimGalore, alignment using BWA-MEM, processing of the BAM file using SAMtools, and variant calling using FreeBayes. Then, the generated VCF file is used as input to the R script for the resistance prediction using the ML models and prediction explanation using SHAP. For the BDQ resistance prediction from VCF files, the scripts are similar to those of the web application. In this standalone application, the prediction output and probabilities generated by the R script are stored in TSV files, and SHAP plots are, in addition, created and saved in Scalable Vector Graphics (SVG) format.

## Data Availability

The Supplementary files provide the supporting figures and the tables. The data are available in the BDQR-MTB web application http://bicresources.jcbose.ac.in/ssaha4/bdqr-mtb and the standalone version BDQR-MTB standalone https://github.com/PulmonomicsLab/bdqr-mtb-standalone.

## Supplementary Information

## Acknowledgement

The authors acknowledge the core computing facility of Bose Institute, and S Ghosh acknowledges University Grants Commission (UGC) for the fellowship. The authors acknowledge Dr. Abhirupa Ghosh for writing the code for the variant calling pipeline used in this work. This work was supported by the Bioinformatics center of Bose Institute, funded by the Department of Biotechnology, Government of India, vide the sanction no. BT/PR40174/BTIS/137/45/2022.

### Author Contributions

S Ghosh: data curation, formal analysis, methodology, and writing—original draft.
S Bhattacharjee: software and visualization.
S Saha: conceptualization, formal analysis, supervision, investigation, and writing—review and editing.

### Conflict of Interest Statement

The authors declare that they have no conflict of interest.

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
