## [Reviewer comments · Life Science Alliance]

Machine learning method for the prediction of Bedaquiline-resistant *Mycobacterium tuberculosis*

Stuti Ghosh, Sudipto Bhattacharjee, and Sudipto Saha
DOI: <https://doi.org/10.26508/lsa.202503539>

Corresponding author(s): Sudipto Saha, Bose Institute

Review Timeline:

Submission Date:	2025-10-16
Editorial Decision:	2026-01-02
Revision Received:	2026-01-24
Editorial Decision:	2026-03-16
Revision Received:	2026-03-23
Accepted:	2026-04-15

Scientific Editor: Sarita Hebbar

Transaction Report:

January 2, 2026

Re: Life Science Alliance manuscript #LSA-2025-03539-T

Dr. Sudipto Saha
Bose Institute

Dear Dr. Saha,

Thank you for submitting your manuscript entitled "Machine learning method for the accurate identification of Bedaquiline-resistant Mycobacterium tuberculosis clinical isolates and their novel drug resistance genes from whole genome sequencing" to Life Science Alliance. We apologise for the delay in communicating our decision due to editor availability issues and previous delays in securing reviewer comments.

The manuscript was assessed by two expert reviewers, whose comments are appended to this letter. As you will note, both reviewers agree that your work on a BDQ-specific predictive tool is of interest to the community. That said, they also raised major concerns that preclude publication at this stage.

We are particularly grateful to Reviewer 1 for highlighting an important concern, that the use of a stringent threshold (in the absence of any rationale) may bias the analyses towards high coverage genomic areas whilst excluding clinically relevant mutations. We agree with this reviewer that you must include (1) a rationale for using a read depth threshold greater than 150 and (2) a sensitivity analysis, and (3) an elaboration on the potential bias of this stringent threshold. We also agree with both reviewers that you must provide all missing information (as indicated in their comments/questions). Finally we concur with Reviewer 1 that you must substantially modify the methods sections to contain all the pertinent information and not refer to the figures for details.

In line with the overall comments, we invite you to submit a revised manuscript addressing the Reviewer comments.

I would be happy to discuss the revision in more detail via email or phone/videoconferencing. Please let me know which option you prefer, if any.

While you are revising your manuscript, please also attend to the below editorial points to help expedite the publication of your manuscript. Please direct any editorial questions to the journal office. When submitting the revision, please include a letter addressing the reviewers' comments point by point.

Thank you for this interesting contribution to Life Science Alliance. We hope that the comments below will prove constructive as your work progresses, and we are looking forward to receiving your revised manuscript.

Best wishes for the new year,

Sarita Hebbar, PhD
Scientific Editor
Life Science Alliance
<http://www.lsajournal.org>

- A letter addressing the reviewers' comments point by point.
- An editable version of the final text (.DOC or .DOCX) is needed for copyediting (no PDFs).

B. MANUSCRIPT ORGANIZATION AND FORMATTING:

Reviewer #1 (Comments to the Authors (Required)):

This manuscript addresses a timely and important question: the prediction of bedaquiline resistance using machine learning and whole-genome sequencing. The approach is promising, and the potential development of a BDQ-specific predictive tool is of significant interest to the TB genomics community.

However, the manuscript requires major revisions to improve:

Methodology Section

The section reads more like a workflow description rather than a structured Methods section. It constantly refers to Figures 4A-C, but much of this information should be in the text, not outsourced to figures.

Variant filtering thresholds

A read depth threshold greater than 150 is unusually stringent and may remove true mutations from lower-depth regions, bias the model toward genomic areas with high coverage, reduce the detection of low-frequency variants, and exclude clinically relevant heteroresistant mutations. Because no justification is provided for this cutoff, the authors should explain the rationale behind selecting such a high threshold and discuss the potential biases it introduces, ideally supported by a sensitivity analysis.

Missing key data information:

What proportion was used for training vs. testing?

Was stratified sampling used to preserve BDQ-R class balance?

Were BioProjects split across train/test or mixed? (critical to avoid data leakage)

Were multiple random splits evaluated?

Reviewer #2 (Comments to the Authors (Required)):

The drug-resistance of BDQ is very urgent matter. The author employed machine learning (ML) approaches to predict BDQ resistance in Mycobacterium tuberculosis (M. tuberculosis) clinical isolates based on mutations identified from whole-genome sequencing (WGS) data. This is a very meaning event. I have some questions to ask author:

1. Whether do the data contain the people of different races?

2. Whether do the ML model demonstrated optimal performance in predicting BDQ resistance, and XAI enabled us to identify the top resistance associated features to all the races?

Rebuttal

We thank the reviewers for their favorable comments and suggestions, which helped us strengthen the manuscript. In the revised manuscript, changes have been made in the method section write-up, and a sensitivity analysis was performed for four different read depth (DP) thresholds. We provided the Bio-Projects IDs with country-specific information and tested our standalone model on a small dataset from four countries.

Reviewer comments

Reviewer # 1

- 1) This manuscript addresses a timely and important question: the prediction of bedaquiline resistance using machine learning and whole-genome sequencing. The approach is promising, and the potential development of a BDQ-specific predictive tool is of significant interest to the TB genomics community.

However, the manuscript requires major revisions to improve:

Methodology Section

Query 1: The section reads more like a workflow description rather than a structured Methods section. It constantly refers to Figures 4A-C, but much of the information should be in the text, not outsourced to figures. :

Reply 1: We thank the reviewer for the positive comments and suggestions. As suggested, in the revised manuscript, we have structured the method section (pages 15-19)

Variant filtering thresholds

Query 2: A read depth threshold greater than 150 is unusually stringent and may remove true mutations from lower-depth regions, bias the model toward genomic areas with high coverage, reduce the detection of low-frequency variants, and exclude clinically relevant heteroresistant mutations. Because no justification is provided for this cutoff, the authors should explain the rationale behind selecting such a high threshold and discuss the potential biases it introduces, ideally supported by a sensitivity analysis:

Reply 2: We thank the reviewer for this comment. As suggested, we used four different read depth thresholds to evaluate the performance of the ML models, as shown in Table S1

of Supplementary_file_1, where a read depth of 150 was found to be optimal based on sensitivity and computational cost (page 7, line nos, 127-129).

Missing key data information:

Query 3: What proportion was used for training vs. testing?

Reply 3: The dataset was randomly split into an 80:20 proportion. 80% of the data was used for training the machine learning (ML) model, with 5-fold cross-validation. The remaining 20% data was used for blind testing. (Page no. 16, line nos. 329-342).

Query 4: Was stratified sampling used to preserve BDQ-R class balance?

Reply 4: No.

Query 5: Were Bio Projects split across train/test or mixed? (critical to avoid data leakage)

Reply 5: The bio-projects sample runs were mixed and randomly sorted.

Query 6: Were multiple random splits evaluated?

Reply 6: We performed a 5-fold CV, and the performance measures are the averages across 5 models.

Reviewer #2 (Comments to the Authors (Required):

The drug-resistance of BDQ is very urgent matter. The author employed machine learning (ML) approaches to predict BDQ resistance in Mycobacterium tuberculosis (M. tuberculosis) clinical isolates based on mutations identified from whole-genome sequencing (WGS) data. This is a very meaningful event. I have some questions to ask author:

Query1. Whether do the data contain the people of different races?

Answer 1: We thank the reviewer for the positive comments. As suggested, in the revised manuscript, the details of the BioProject IDs, including country information, are provided on page no. 16, line 334-335 (Table S9 in Supplementary_file_1).

Query 2. Whether do the ML model demonstrated optimal performance in predicting BDQ resistance, and XAI enabled us to identify the top resistance associated features to all the races?

Reply 2: As suggested by the reviewer, the optimal ML model used on the web server was tested on a small dataset of 21 MTB clinical isolates from four countries and XAI enabled drug-resistant gene locus obtained as shown in Table S8 in Supplementary File 1 (Page no. 11, Line nos.. 229-232).

March 16, 2026

RE: Life Science Alliance Manuscript #LSA-2025-03539-TR

Dr. Sudipto Saha
Bose Institute
EN-80, Sector-V, Bidhan Nagar,
Kolkata-, West Bengal, 700091
India

Dear Dr. Saha,

Thank you for submitting your revised manuscript entitled "Machine learning method for the prediction of Bedaquiline-resistant Mycobacterium tuberculosis".

Your manuscript was reviewed by one of the original reviewers (Reviewer 2) whose comments are appended below. As you will read, Reviewer 2 has commented favourably on your revised manuscript. For concerns from Reviewer 1, we request you to ensure that your responses are reflected with any necessary text changes in the manuscript to clarify these points.

Overall, we would be happy to publish your paper in Life Science Alliance pending resolution of the above point and final revisions necessary to meet our formatting guidelines. We request you to submit a revised manuscript document with all these changes highlighted

MANUSCRIPT ORGANIZATION AND FORMATTING:

To avoid unnecessary delays in the acceptance and publication of your paper, please read the following information carefully. Full guidelines are available on our Instructions for Authors page, <https://www.life-science-alliance.org/authors>

- Please restructure the legends for main and supplementary figures such that they have a heading, and are not redundant with the Results section in alignment with LSA's guidelines (<https://www.life-science-alliance.org/manuscript-prep#legends>).
- Thank you for providing a Data Availability section. Please ensure that this GitHub page/link has all relevant source code(s) for readers to understand and replicate your findings, in addition to providing access to the standalone BDQR-MTB explainable AI model. For your web-based tool/platform, we encourage you to host your website at a stable, external location rather than through your institution. Should you choose to eventually relocate your platform to such an external web host, we can make this update to your published manuscript (upon request).
- Please revisit your methods sections to completely describe all the methods. For example, please elaborate on the inclusion and exclusion criteria "Certain inclusion and exclusion criteria were used to select the datasets for the current study was performed using classical algorithms such as Random Forest (RF), support vector machine (SVM) and deep learning (DL) based MLP algorithms".
- Please go through your text and carefully call out figures/supplementary items wherever appropriate. For example in the methods section, line 318: please refer to the appropriate table with 16 BioProject IDs.
- We encourage you to make the sub-headings of the results section more informative.
- Please change the call-out to Fig 4 (A-C), line 365 since this figure has no sub-panels labelled as A, B, or C.
- Please provide a citation for the 'Proksee' server mentioned in line 194.
- Please remove the separate supporting information file. Supplementary figures should be uploaded like main figures, separately and individually.
- Please add your table legends to the main manuscript text after the legends for figures.
- Please add the X and Bluesky handles of your host institute/organization, as well as your own, and/or one of the authors, in our system.
- The titles in both the system and the manuscript file must be consistent with each other.
- Please remove the title, lists of the authors and affiliations from the second page in the manuscript file. There is no need to duplicate this information.
- Please rename "Availability of data and materials" to "Data Availability."
- Please add an Author Contributions section to your main manuscript text.
- Please be sure that the authorship listing and order is correct.

per figure for this information. These files will be linked as supplementary "Source Data" files.

We welcome submissions of potential cover images for the issue of LSA in which your work would appear. If you have high quality images associated with this work, please feel free to email these, with a caption, to the journal office.

LSA encourages authors to provide a 30-60 second video where the study is briefly explained. We will use these videos on social media to promote the published paper and the presenting author (for examples, see <https://docs.google.com/document/d/1-UWCfbE4pGcDdcgzcmiuJI2XMBJnxKYeqRvLLrLSo8s/edit?usp=sharing>). Corresponding or first-authors are welcome to submit the video. Please submit only one video per manuscript. The video can be emailed to contact@life-science-alliance.org

FINAL FILES:

The following items are required for acceptance.

The license to publish form must be signed before your manuscript can be sent to production. A link to the license to publish form will be available to the corresponding author only. Please take a moment to check your funder requirements.

Thank you for your attention to these final processing requirements. Please revise and format the manuscript and upload materials as soon as you are able.

Thank you for this interesting contribution to the literature. We look forward to publishing your paper in Life Science Alliance.

Sincerely,

Sarita Hebbar, PhD
Scientific Editor
Life Science Alliance
<http://www.lsajournal.org>

Reviewer #2 (Comments to the Authors (Required)):

The article addresses the increasing resistance to the FDA-approved drug Bedaquiline (BDQ) in Mycobacterium tuberculosis (MTB). The revision of manuscript is suitable for publication.

Additional changes done in the manuscript (Revision)**Correction in the email ID of one author**

- 1) **Page – 1, line no 6:** The email ID of one of the authors was wrong. The email ID of author Sudipto Bhattacharjee was corrected. The email ID was initially mentioned as tsuditpto@gmail.com. The corrected email ID is: ttsudipto@gmail.com.

Abbreviations were introduced

- 2) **Page – 2, line no 34:** The full form of SHAP was not mentioned previously. It was introduced.
- 3) **Page – 3, line no 39:** The full form of XAI was not mentioned in the keywords section. It was introduced.
- 4) **Page – 4, line no 42:** The term TB was introduced. So the full form was mentioned.
- 5) **Page – 5, line no 71:** The full form of ARGs was mentioned as it was not previously done in the introduction part.
- 6) **Page – 5, line no 74:** The full form of DST was not mentioned previously. The full form of DST was introduced.
- 7) **Page – 5, line no 78:** The full form of MDR-TB was already mentioned previously. So the abbreviation was mentioned.

Corrections of the grammatical errors

- 8) **Page – 2, line no 18:** The name of the pathogen was not italicized. It was changed to italics.
- 9) **Page – 2, line no 27:** There was no space between variant calling format and (VCF). The space was introduced.
- 10) **Page – 4, line no 55:** The rising incidence of bedaquiline-resistant strains presents a severe challenge to the global health community in managing tuberculosis, as it jeopardises the efficacy of one of the essential possibilities for the treatment of severe tuberculosis. The last part of the line from 'as it jeopardises' was removed because it added no extra meaning in the introduction section.
- 11) **Page – 9, line no – 158:** There was a typographical error. The dataset name was wrongly mentioned as $\text{top}_{50}\text{RF_BDQR}_{\text{high}}$. It was corrected to $\text{top}_{50}\text{RF_BDQR}^{\text{high}}$.

Corrections of errors in the number of samples

- 12) The number of BDQ-susceptible samples in the dataset was mentioned as 340 (page 6, line no. 116) and as 338 (page 15, line no 318) in the previous version. This was corrected to 338 in **page 6, line 99** of the current version (revision 2).
- 13) **Page – 6, line no – 100:** The number of samples considered for ML training with 5-fold cross-validation was wrongly mentioned as 506. It was changed to 507.

Minor additions and changes were made to the text for clarity in the revised version. It didn't reflect any changes in the results and conclusion of the manuscript.

- 14) **Page – 8, line no – 138:** The line was corrected. It was previously mentioned that the feature selection is 'iterative', meaning repetitive. This was corrected as the feature

selection was performed by using VarImp function based on variable importance scores. Moreover, the different feature numbers used in the ROC plots (Figure 1d) were not previously reported. So the line was corrected to mention the different feature numbers considered.

- 15) **Page – 12, line no – 226 to 227:** The line was corrected as the results are based on predictions. It was previously mentioned as ‘Thus, the features selected based on the RF model were concluded to be more biologically significant’.
- 16) **Page – 16, line no – 331:** The subheading was previously machine learning. It was changed to introduce the SHAP based explainability part with machine learning. This was done because the subheading "machine learning" is very general.
- 17) **Page -17, line no – 346 to 350:** The different number of features considered for ML model training in order to select the minimalistic model was not mentioned previously in the methods section. This information was introduced.
- 18) The machine learning model development subheading (page -17, line no: 366 of the previous version) was removed. The part was mentioned under the subheading ‘Machine learning pipeline and SHAP-based explainability for BDQ resistance prediction’ (page 16 and line no 331).
- 19) **Page – 22, line no – 447 to 449:** The funding information was incorporated within the acknowledgement section.
- 20) **Page – 29, line no – 697:** It was previously mentioned that the blind testing was done on an independent dataset. This was corrected as the testing was done on the blind dataset, which is 20% of the dataset. The number of training samples for the 5-fold CV was corrected to 507.

Figure modification

- 21) Figure 4a: The number of BDQ-susceptible samples was corrected to 338.

Reformatting of the supplementary tables

- 22) The supplementary tables were provided in .xls format according to the journal pattern. They were mentioned as **table S1, table S2, table S3, table S4, table S5, table S6, table S7, table S8, table S9** and **table S10**.

We thank the reviewer for suggestions and comments. Here, we have answered all the queries raised.

1)-Please restructure the legends for main and supplementary figures such that they have a heading, and are not redundant with the Results section in alignment with LSA's guidelines (<https://www.life-science-alliance.org/manuscript-prep#legends>).

Response 1: We thank the reviewer for the comment. The figure legends have been modified in accordance with LSA guidelines. The figure legends are provided on pages 30 to 31 of the main manuscript.

2) -Thank you for providing a Data Availability section. Please ensure that this GitHub page/link has all relevant source code(s) for readers to understand and replicate your findings, in addition to providing access to the standalone BDQR-MTB explainable AI model. For your web-based tool/platform, we encourage you to host your website at a stable, external location rather than through your institution. Should you choose to eventually relocate your platform to such an external web host, we can make this update to your published manuscript (upon request).

Response 2: All the relevant source code(s) have been provided in the GitHub page/link. We don't have access to any external web host.

3) -Please revisit your methods sections to completely describe all the methods. For example, please elaborate on the inclusion and exclusion criteria "Certain inclusion and exclusion criteria were used to select the datasets for the current study was performed using classical algorithms such as Random Forest (RF), support vector machine (SVM) and deep learning (DL) based MLP algorithms".

Response 3: The above-mentioned sections were elaborated and rewritten. The changes were incorporated in the manuscript (page 18, lines 300 to 309 and page 20, lines 338 to 345).

4) -Please go through your text and carefully call out figures/supplementary items wherever appropriate. For example in the methods section, line 318: please refer to the appropriate table with 16 BioProject IDs.

Response 4: We are extremely thankful to the reviewer for this comment. The bioproject ID table (S9) contains 17 IDs. The number of bioproject IDs considered for the study were 17. The changes were incorporated into the main manuscript (page 18, line 299) and figure 4A.

5) -We encourage you to make the sub-headings of the results section more informative.

Response 5: We thank the reviewer for the suggestion. The subheadings of the results section were modified (page 6: line 98; page 7: line 110; page 9: line 146; page 12: line 199).

6) -Please change the call-out to Fig 4 (A-C), line 365 since this figure has no sub-panels labelled as A, B, or C.

Response 6: We thank the reviewer for this point. The figure 4 was modified and the sub panels were renamed as 4 (a-c).

7) -Please provide a citation for the 'Proksee' server mentioned in line 194.

Response 7: The citation for the 'Proksee' server was added to the main manuscript (page 11: line number 177).

8) -Please remove the separate supporting information file. Supplementary figures should be uploaded like main figures, separately and individually.

Response8: The separate supporting information file was removed. The supplementary figures were cited as Fig S1(line no: 134), Fig S2 (line number: 136), Fig S3 (line number: 153), Fig S4 (line number: 160), fig S5(line number: 169), Fig S6 (line number: 421-422) respectively.

9) -Please add your table legends to the main manuscript text after the legends for figures.

Response 9: The table legends were added to the main manuscript following the figure legends (line numbers 673 to 698).

10) -Please add the X and Bluesky handles of your host institute/organization, as well as your own, and/or one of the authors, in our system.

Response9: Our institute has an X account; once published, we will add it to it.

11) -The titles in both the system and the manuscript file must be consistent with each other.

Response 11: Done

12) -Please remove the title, lists of the authors and affiliations from the second page in the manuscript file. There is no need to duplicate this information.

Response 12: The above-mentioned modification has been done in page number 2 of the manuscript.

13) -Please rename "Availability of data and materials" to "Data Availability."

Response 13: This change has been incorporated in the manuscript (page 24: line number 434).

14) -Please add an Author Contributions section to your main manuscript text.

Response 14: The Author Contributions section was added to the manuscript (line number : 438).

15) -Please be sure that the authorship listing and order is correct.

Response 15: The authorship listing was checked and found to be correct.

April 15, 2026

RE: Life Science Alliance Manuscript #LSA-2025-03539-TRR

Dr. Sudipto Saha
Bose Institute
EN-80, Sector-V, Bidhan Nagar,
Kolkata-, West Bengal, 700091
India

Dear Dr. Saha,

Thank you for submitting your Research Article entitled "Machine learning method for the prediction of Bedaquiline-resistant Mycobacterium tuberculosis". It is a pleasure to let you know that your manuscript is now accepted for publication in Life Science Alliance. Congratulations on this interesting work. We appreciate your confirming the required changes to manuscript files that the editors and reviewers requested, and we regret the delay that this caused in issuing a final decision on this work.

DISTRIBUTION OF MATERIALS:

Again, congratulations on a very nice paper. I hope you found the review process to be constructive and are pleased with how the manuscript was handled editorially despite the delays. We look forward to future exciting submissions from your lab.

Sincerely,

Tim Fessenden, PhD
Scientific Editor
Life Science Alliance
<http://www.lsajournal.org>